# Predicting multiple sclerosis from radiologically isolated syndrome using generative artificial intelligence

Christine Lebrun-Frenay[1,2*], Felix Renard[3], Lydiane Mondot[1,2], Cassandre Landes-Chateau[1], Adeline Stewart[3], Mikael Cohen[1,2], Darin T. Okuda[4,5], Arnaud Attyé[3]

**1** UR2CA-URRIS, Université Nice Côte d'Azur, Nice, France, **2** CRCSEP Nice, Département de Neurologie CHU de Nice Pasteur 2, Nice, France, **3** GeodAIsics. Biopolis - La Tronche, France, **4** Department of Neurology, Neuroinnovation Program and Multiple Sclerosis and Neuroimmunology Imaging Program, The University of Texas Southwestern Medical Center, Dallas, Texas, United States of America, **5** Peter O'Donnell Brain Institute, The University of Texas Southwestern Medical Center, Dallas, Texas, United States of America

* lebrun-frenay.c@chu-nice.fr

## Abstract

Radiologically Isolated Syndrome (RIS) is characterized by incidental MRI findings indicative of multiple sclerosis (MS) in asymptomatic individuals. Factors such as younger age, positive cerebrospinal fluid biomarkers, and specific lesion locations have been previously linked to a higher risk of conversion from RIS to clinical MS. Predicting which individuals will develop clinical MS remains challenging. Based on widely available cross-sectional patient studies, unsupervised machine learning has been proposed to uncover MRI-driven MS phenotypes with distinct temporal progression patterns. We evaluated whether an unsupervised artificial intelligence framework based on generative manifold learning could stratify RIS patients by conversion risk. BrainGML-MS analyzed imaging biomarkers and generated individualized digital twins from MRI data. We studied 152 RIS individuals (32 converters, RIS-C), 152 MS patients, and 152 healthy controls. The model identified four RIS clusters with distinct five-year conversion risks ranging from 10% to 39%. The brain age gap increased progressively from healthy controls to RIS non-converters, RIS-C, and MS. RIS converters showed greater structural atrophy and greater similarity to MS profiles. These findings indicate that MRI-derived brain aging biomarkers and structural deviations measured at the first RIS scan may improve early risk stratification and support clinical decision-making in preclinical MS.

## Author summary

We developed an artificial intelligence method that analyzes brain MRI scans at diagnosis to identify people with radiologically isolated syndrome who are

---

---

**Data availability statement:** We used 3D T1-weighted MRI data from nine publicly available databases covering the entire lifespan. https://github.com/volBrain/AssemblyNet. The study protocol is available on protocols.io (DOI: dx.doi.org/10.17504/protocols.io.kxygx826zv8j/v1). Reduced-space manifold coordinates, clustering outputs, and associated clinical metadata are available in the companion GitHub repository: https://github.com/Geodaisics-code/ScoRIS. For proprietary reasons, the code for BrainGML-MS is not publicly available. While the full BrainGML-MS software is not publicly distributed due to regulatory and proprietary constraints, all preprocessing steps, feature definitions, and model parameters are described in sufficient detail to enable independent methodological replication. Derived data supporting the findings are available from the corresponding author upon reasonable request.

**Funding:** The author(s) received no specific funding for this work.

**Competing interests:** I have read the journal's policy and the authors of this manuscript have the following competing interests: Christine Lebrun-Frenay has no conflict of interest. Félix Renard is co-founder and stockholder of GeodAIsics. Lydiane Mondot has received personal compensation for consulting, speaking, or other activities with Biogen, Sandoz, Alexion. Cassandre Landes-Château has no conflict of interest. Adeline Stewart is employed by GeodAIsics. Mikael Cohen has received personal compensation for consulting, serving on a scientific advisory board, speaking, or other activities with Biogen, Merck, Sanofi, Roche, Celgene-BMS, Janssen, Alexion, Horizon Therapeutics, and Ad Scientiam. Darin T Okuda received personal compensation for consulting and advisory services from Biogen Inc., Cortechs.AI, Eisai, EMD Serono, Genentech, Genzyme/Sanofi, Immunic Therapeutics, Moderna, RVL Pharmaceuticals, Inc., and Zenas BioPharma along with research support from Alexion, EMD Serono/Merck and Novartis. D.T.O. has issued national and international patents and pending patents related to other developed technologies. D.T.O. received royalties for patents related to other developed technologies. D.T.O. received royalties for intellectual property licensed by The Board of Regents of The University of Texas System. D.T.O. is the Founder of Revert Health Inc. Arnaud Attyé is a co-founder and stockholder of GeodAIsics.

more likely to develop clinical multiple sclerosis. The system compares each RIS individual to a personalized "digital twin" representing the expected healthy brain structure. We found that specific patterns of brain aging and structural changes on the first RIS MRI scan were associated with a higher risk of clinical disease onset. This approach could help clinicians identify RIS individuals who may benefit from closer monitoring or early treatment.

## Introduction

Radiologically isolated syndrome (RIS) is defined by incidental MRI findings suggestive of multiple sclerosis (MS) in individuals without clinical symptoms [1,2]. Although many individuals with RIS remain asymptomatic, a substantial proportion later develop clinical MS, creating uncertainty in clinical management [3]. Previously published data by the International Consortium for Research on RIS (RISC) involving the largest cohort of participants monitored after their first brain MRI suggestive of MS revealed that 13.8% experienced a first clinical event (FCE) defining clinical MS within two years [4], 34% within five years [5], and 51.2% within ten years [6]. Existing predictors of conversion risk, including age, cerebrospinal fluid biomarkers, and lesion location, provide limited individual-level accuracy [1,7]. No validated risk score or imaging biomarker currently enables reliable prediction of clinical conversion.

Quantitative MRI has become essential for detecting subtle brain abnormalities by comparing individual morphometric values to normative reference distributions. Recently, the SuStaIn framework applied unsupervised artificial intelligence (AI) to stratify patients with neurodegenerative diseases into subtypes based on pseudo-temporal disease progression [8,9]. Although appearing promising, this model showed strong associations with disease severity and baseline impairment in MS patients but lacked predictive power for future disability [10].

As an alternative, Generative Manifold Learning (GML) was introduced to create individualized "digital twins," enabling personalized normative modeling using manifold-learning techniques [11]. Unlike traditional approaches that rely on population averages, GML captures the intrinsic geometry of unsupervised MRI data, enabling patient-specific comparisons. This framework offers both interpretability and clinical relevance, particularly in the preclinical stages of neurodegenerative diseases, such as RIS for MS. A generative component reconstructs a subject-specific "healthy" reference, enabling deviation analysis using morphometric biomarkers [12,13]. Among these, the Brain Age Gap (BAG)—the difference between estimated brain age and chronological age—has already been reported as relevant in MS [14]. Advanced brain age appears to be correlated with long-term disability, lesion burden, and disease phenotype, with an overestimation of brain age in the progressive form compared with the relapsing course.

The BrainGML-MS is a software platform developed by GeodAIsics (www.geodaisics.com) that supports radiologic assessment of patients with neurological disorders. It has already demonstrated its value in epilepsy and dementia related disorders [12].

This study evaluates the BrainGML-MS BAG's ability to support an interpretable, clinically actionable model that bridges the preclinical-to-clinical MS continuum and improves early risk stratification to inform the development of clinical symptoms. To achieve this, we applied the unsupervised learning GML-based approach to a cohort of RIS individuals and MS patients. Our aim was to identify radiologically defined subgroups within the preclinical phase, assess their risk of clinical conversion, and describe their imaging profiles from their initial diagnostic brain MRI scan.

## Results

### Cohort characteristics

A total of 152 individuals with RIS were included (77% female; mean age 41.2 ± 13.8 years). Among them, 120 remained asymptomatic during follow-up (RIS-nonconverters; RIS-NC; mean age 41.8 ± 13.3 years), and 32 experienced a first clinical event (RIS-converters; RIS-C; mean age 35.7 ± 12.5 years). None of the individuals who converted during follow-up developed a primary progressive multiple sclerosis (PPMS) phenotype. The study also included 152 matched healthy controls (HC) and 152 matched patients with MS. Spinal cord MRI and cerebrospinal fluid analyses were available for 94 RIS individuals (61.8%). Mean follow-up duration for the RIS cohort was 3.3 ± 3.5 years (Table 1).

### Brain age gap differences between RIS and other groups

The brain age gap differed significantly across groups (Kruskal–Wallis, $p < 0.001$). Healthy controls showed predominantly negative BAG values, indicating younger-appearing brains, whereas MS patients showed positive BAG values consistent with older-appearing brains. RIS individuals fell between these extremes. Within RIS, converters exhibited higher BAG values than non-converters, placing them closer to MS distributions. Overall, BAG demonstrated a progressive shift from HC to RIS-NC to RIS-C to MS, reflecting increasing structural deviation along the disease continuum (Fig 1).

### Regional structural brain differences

The number of regions with significant deviation ($|z| > 1.96$) increased across groups: HC (mean 6.8 regions), RIS-NC (14.4), RIS-C (15.6), and MS (16.3). Spatial maps showed that the proportion of subjects with abnormal regions rose progressively from HC to MS. The third ventricle was the most discriminative structure, distinguishing HC from all other groups, suggesting that early ventricular enlargement may be a marker of disease-related change. Overall, these findings indicate a graded increase in structural brain abnormalities across disease stages (Fig 2; Table 2).

### *Assessing clinical conversion risk in RIS individuals and the influence of associated biomarkers*

Unsupervised clustering identified four RIS subgroups with distinct conversion risks ranging from 10% to 39% over five years. Cluster 4 contained a significantly higher proportion of converters than clusters 1 and 2 (Fisher's exact test $p < 0.05$). Cluster 4 also overlapped more strongly with MS distributions, whereas cluster 2 showed greater similarity to HC profiles. Among evaluated biomarkers, only spinal cord lesions differed significantly between clusters ($p = 0.04$), occurring more frequently in clusters 2 and 3. Other markers, including T2 lesion load and CSF positivity, did not differ significantly.

**Table 1. RIS MRI index scan characteristics (lesions' anatomical location according to 2017 MS criteria [15]), and CSF positivity (presence of at least two unique oligoclonal bands).**

|  | Spinal Cord Lesions | Infratentorial Lesions | Gadolinium enhancing-lesions | CSF positive |
|---|---|---|---|---|
| Yes | 31 | 20 | 11 | 68 |
| No | 63 | 74 | 80 | 26 |

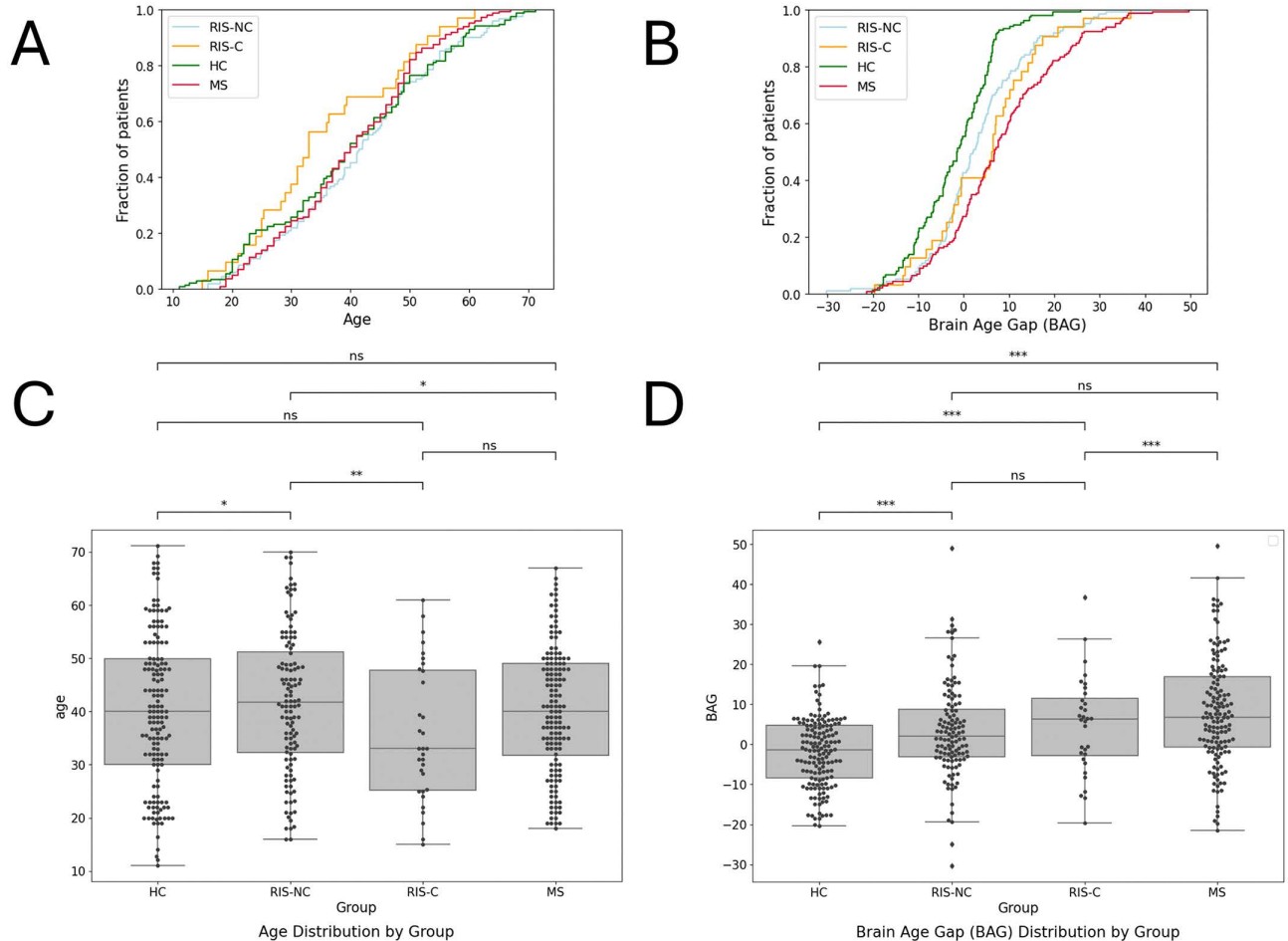

**Fig 1. Distribution of age and brain age gap across study groups.** Empirical cumulative distribution plots (A–B) and boxplots (C–D) showing chronological age and brain age gap (BAG) across healthy controls (HC), radiologically isolated syndrome non-converters (RIS-NC), RIS converters (RIS-C), and multiple sclerosis (MS). BAG represents the difference between predicted brain age and chronological age. Group differences were assessed using Kruskal–Wallis tests with significance thresholds indicated as ns ($p > 0.05$), * ($p \le 0.05$), ** ($p \le 0.01$), *** ($p \le 0.001$). These distributions demonstrate a progressive shift in BAG from HC to RIS-NC to RIS-C to MS, indicating increasing structural deviation along the disease continuum.

These findings indicate that cluster membership stratifies RIS individuals by conversion risk and structural similarity to MS (Fig 3; Table 3). Cluster membership remained associated with conversion status after accounting for age, sex, lesion load, and spinal cord lesion presence in multinomial regression analysis.

**Trajectory modeling**

Trajectory analysis along the manifold progression axis revealed distinct temporal patterns of regional abnormalities (Fig 4). RIS converters and non-converters showed different timing of structural deviations, particularly in ventricular regions. For example, all individuals in cluster 1 exhibited significant deviation in the left inferior lateral ventricle at approximately 24 months, whereas similar changes appeared only around 60 months in cluster 4. These results indicate that structural abnormalities emerge earlier and progress more rapidly in converter-like profiles, supporting distinct disease-evolution pathways.

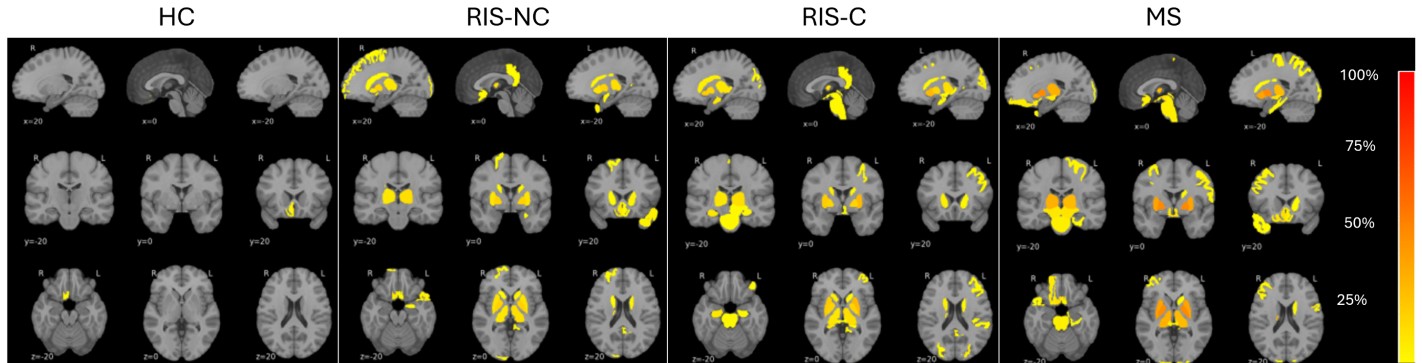

**Fig 2. Regional brain abnormality burden across groups.** Sagittal, coronal, and axial maps showing the percentage of subjects within each group who exhibited significant regional deviations (|z| > 1.96) for each region of interest (ROI). Z-scores were computed relative to individualized digital twin references. Percentages indicate the number of subjects exceeding the significance thresholds in each ROI. Warmer colors indicate a higher proportion of abnormal regions within a group. The increasing number of abnormal regions from HC to RIS-NC to RIS-C to MS demonstrates a graded rise in structural brain alterations across disease stages.

**Table 2. Top 10 ROI in differentiating the different groups and indication of volume increase (in bold) or atrophy (normal font) as the most common in the pair.**

| HC/RIS-NC | HC/MS | HC/RIS-C | RIS-NC/MS | RIS-NC/RIS-C | MS-RIS-C |
|---|---|---|---|---|---|
| **3rd ventricle** | **3rd ventricle** | **3rd ventricle** | Left Pallidum | Left Putamen | Left Cerebellum White Matter |
| Left Cerebral White Matter | Left Cerebellum White Matter | **Left Amygdala** | Left Putamen | **Left sup. occipital gyrus** | Left angular gyrus |
| Left Pallidum | **Left Inf Lat Vent** | Left Pallidum | Left parahippocampal gyrus | Left sup. temporal gyrus | Left precentral gyrus |
| Left Thalamus | Left Pallidum | Left Putamen | Left precentral gyrus | Right Cerebellum White Matter | Lobules VI-VII |
| Right Amygdala | Left Putamen | Left Thalamus | Right Pallidum | Right Putamen | Right Cerebellum White Matter |
| Right Cerebral White Matter | Left precentral gyrus | Right Accumbens | Right Putamen | **Right sup. occipital gyrus** | Right occipital pole |
| Right Pallidum | Right Pallidum | Right Amygdala | **Right gyrus rectus** | Right supramarginal gyrus | Right precentral gyrus |
| Right Thalamus | Right Putamen | Right Pallidum | Right parahippocampal gyrus | Sym lateral orbital gyrus | **Right precentral gyrus medial segment** |
| **Right posterior orbital gyrus** | **Right medial orbital gyrus** | Right Putamen | **Right planum polare** | Sym posterior orbital gyrus | Right supramarginal gyrus |
| Sym posterior orbital gyrus | Right precentral gyrus | Sym Cerebellum Grey Matter | Right precentral gyrus | **Sym sup. parietal lobule** | Sym Cerebellum Grey Matter |

## Discussion

In clinical practice, identifying individuals with RIS who are most likely to develop clinical MS is essential for guiding monitoring and treatment decisions. Established risk factors include younger age, male sex, spinal cord lesions, and cerebrospinal fluid oligoclonal bands [4–6]. However, these predictors do not reliably estimate individual conversion risk, and no validated predictive model currently exists. Updated RIS diagnostic criteria now acknowledge that individuals with fewer T2 lesions but positive CSF findings or evidence of dissemination in time (DIT) on MRI face a similar risk of progressing

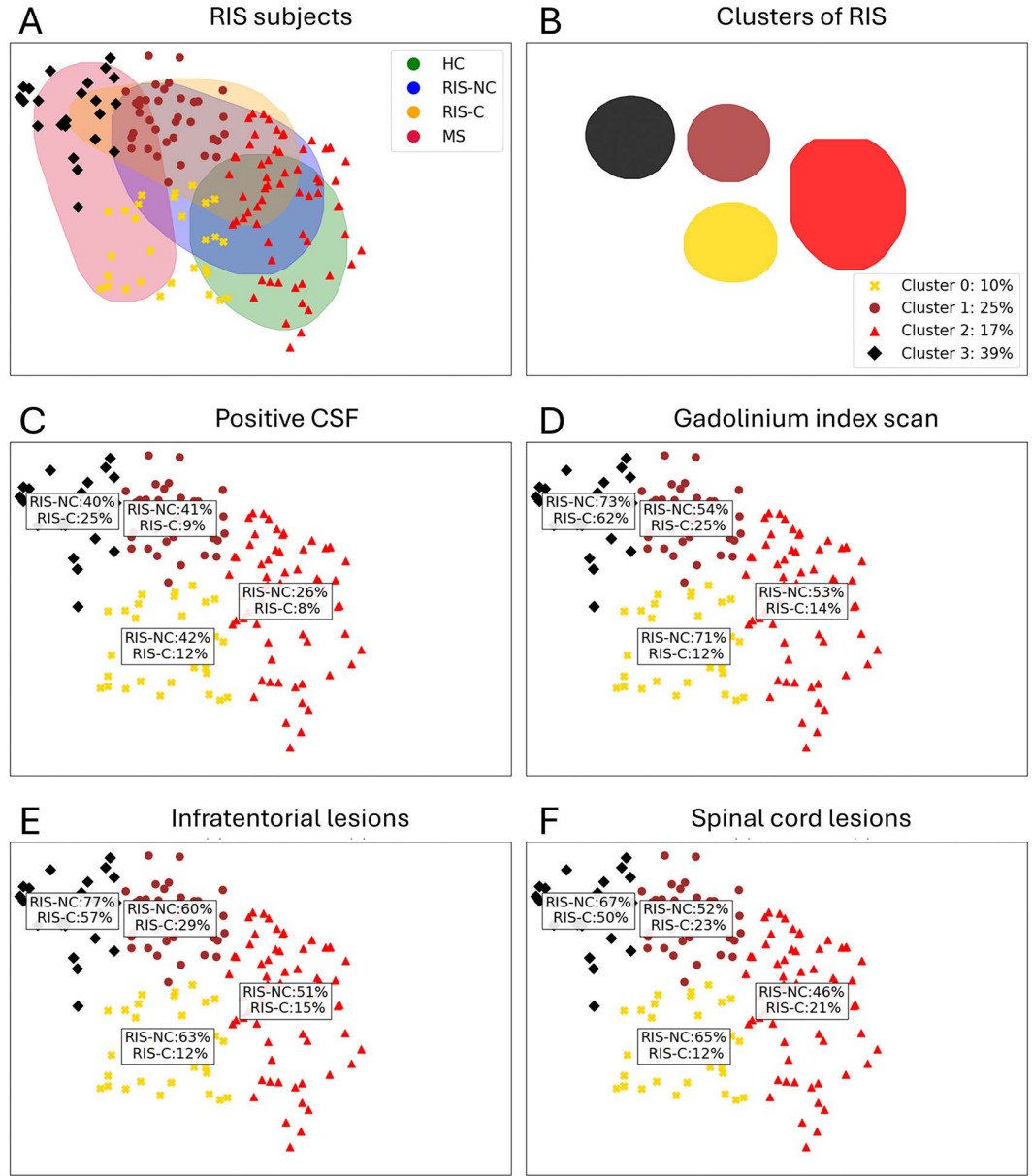

**Fig 3. Cluster distribution and biomarker profiles in RIS.** Spatial distribution of RIS subjects within manifold space and proportion of converters per cluster. **(B)** Density plot of cluster distributions. **(C–F)** Percentages of biomarker-positive individuals within each cluster. RIS subjects were grouped using k-means clustering applied to manifold-embedded data. Clusters were labeled according to conversion risk: red ≥ 25%, orange 10–25%. Differences in converter proportions between clusters were tested using Fisher's exact tests. Cluster membership stratifies RIS individuals by clinical conversion risk, with high-risk clusters showing greater similarity to MS-like structural patterns.

to clinical MS, either with a relapsing or a progressive phenotype [3,17]. Recent advancements, including two phase III randomized placebo-controlled trials in RIS cohorts with disease-modifying therapies prescribed in relapsing MS, have demonstrated that delaying or even preventing the onset of MS is achievable [17,18]. Therefore, the early and accurate identification of RIS patients at high risk of FCE is essential for guiding therapeutic decisions. Despite significant research efforts, no single biomarker or risk score has proven sufficiently reliable to predict progression to clinical MS.

**Table 3. Fisher's exact test p-value for the difference in RIS-C numbers between clusters.**

| cluster | 1 | 2 | 3 |
|---------|------|------|------|
| 1 | 0.19 | | |
| 2 | 0.53 | 0.43 | |
| 3 | 0.02 | 0.26 | 0.04 |

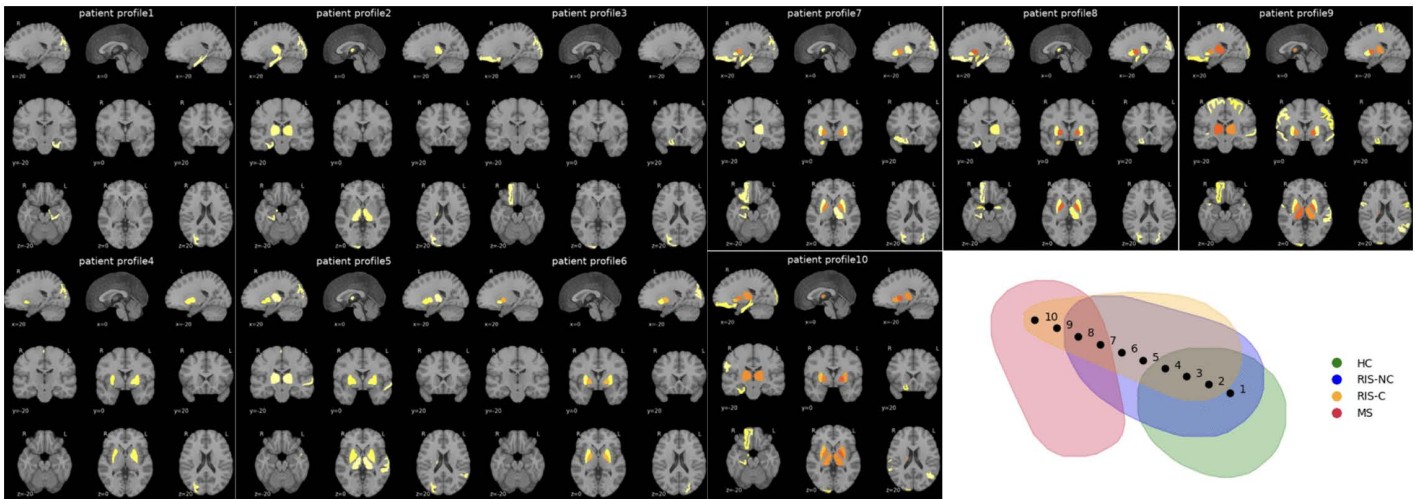

**Fig 4. Representative trajectory of structural progression.** Z-score maps across ten sequential points along a manifold trajectory representing progression from healthy control–like profiles (point 1) to MS-like profiles (point 10). Each point includes sagittal, coronal, and axial views. Trajectories were generated from geodesic distances on the manifold. Significant deviations are color-coded: yellow ($|z| = 1.96$), orange ($|z| \approx 2.3$), red ($|z| \geq 3$). Earlier appearance and expansion of abnormal regions indicate faster progression along the disease axis. Structural abnormalities emerge earlier and progress more rapidly in converter-like profiles than in non-converter-like profiles, supporting distinct disease evolution pathways.

There is a strong demand for diagnostic tools that can accurately predict the progression of RIS to clinical MS and identify individuals who should receive disease-modifying treatment during the presymptomatic phase to reduce the risk of symptoms. In our study, BrainGML-MS revealed significant differences in brain volume between RIS-C and RIS-NC, with greater atrophy observed in the RIS-C group. These volumetric differences allowed the calculation of the BAG, which was higher in RIS-C, aligning them more closely with MS patients. In contrast, RIS-NC exhibited BAG values closer to HC, indicating minimal deviation. BAG significantly predicted the clinical conversion status, and the differences between the HC and MS groups were statistically significant ($p = 0.001$-$0.01$). The inclusion of spinal cord lesion status to the BrainGML-MS model confirmed the considerable value of this marker [19].

Our study also identified significant differences in specific brain regions, such as cerebral white matter, thalami, gyri, and cerebellar gray matter, with some structural atrophy more characteristic of RIS-C and MS. These findings suggest possible progression, observable even in RIS-NC patients, despite the short follow-up. Notably, some RIS individuals showed extreme volume values, warranting further examination. Some RIS subjects with high T2 lesion loads did not exhibit clinical conversion, which may reflect compensatory mechanisms or repair. The lesion significance in the cerebral white matter, thalami, several gyri structures, and cerebellar gray matter has already been recognized as a surrogate marker in MS and RIS [20,21].

The BrainGML-MS model provides insights into RIS progression dynamics and identifies RIS at risk of developing clinical MS. Given that about 50% of RIS patients may never convert to MS, current clinical practice favors conservative

management due to the potential risks of early treatment, such as immune system complications. However, the possibility of reclassifying RIS individuals under the new 2024 McDonald MS diagnostic criteria raises critical questions about early intervention [22].

Unsupervised artificial intelligence development shows promise in enhancing clinical and radiological interpretation of patient data and prognosis. However, the practical implementation is challenged by critical pitfalls, notably the availability of biomarkers in clinical practice, population heterogeneity [23], high-dimensional data with numerous variables [24], and a sample size often too small to accommodate the complexity arising from these issues [25]. These challenges frequently introduce biases, particularly concerning overfitting and the inherent opacity of AI methods, which often complicate model reliability.

The BrainGML-MS software differs from traditional digital twins by adopting a patient-specific, generative approach that reconstructs MRI features from 1.5- or 3T scans as if they were unaffected, providing personalized normative references. Although higher field strengths provide an improved signal-to-noise ratio, standardized preprocessing allows reliable volumetric estimation at 1.5T, supporting inclusion of such data in normative modeling. Unlike classical models based on population data, BrainGML-MS integrates biomechanistic learning to model disease progression and ensure clinical explainability. The Manifold Learning method used in this study provides a nonlinear approach for modeling complex, high-dimensional brain data. Unlike other unsupervised methods, which use predefined progression stages and a limited number of brain regions for analysis, our approach employs unsupervised learning to create personalized "digital twins" [26,27] This technique offers a more dynamic, individualized model for predicting disease progression. In our study, the manifold space accurately distinguishes RIS-NC, RIS-C, and MS clusters. It tracks RIS-C trajectories toward MS clusters, enabling clinicians to anticipate disease progression more effectively. Over time, RIS-C patients are increasingly unable to compensate for lesion loads and neurodegenerative processes, aligning them closer to MS profiles. Although overall conversion rates fall within previously reported ranges, the model's contribution lies in stratifying individuals within the RIS population into distinct structural subgroups with different risk levels. This individualized stratification may support risk-adapted monitoring strategies rather than redefining absolute risk thresholds.

The BAG reflects deviations between estimated and chronological brain age, demonstrating a progressive increase from HC to RIS-NC to RIS-C, and then to MS. The third ventricle was the most distinguishing ROI across groups, confirming prior studies' findings that thalamic volume loss in RIS parallels early MS progression [27]. Identifying RIS clusters suggests that RIS-C is more closely aligned with MS subjects and exhibits greater structural damage over time.

Our study has limitations. RIS is a rare condition, and even in our cohort, the largest available, the number of RIS-C cases remains low, and the follow-up duration is relatively short. We also did not have any RIS-C cases that developed a progressive form [16]. We cannot extrapolate that these RIS individuals would not have different trajectories and phenotypes, especially influenced by the presence of spinal cord lesions. We could not assess whether structural features or clustering patterns differentiate conversion phenotypes as previously published for MS [28,29]. Larger cohorts, including progressive-onset cases, will be required to address this question. Another limitation is that the absence of cardiometabolic and lifestyle variables that may influence brain structure was not systematically available and therefore could not be included as covariates [30]. Future studies incorporating these factors will be important for determining their interactions with imaging-derived biomarkers, such as the BAG.

While morphometric biomarkers provide valuable insights, further data collection is needed to refine predictive models, especially with larger RIS-C datasets. Future studies with larger cohorts and longitudinal data will enhance the model's ability to accurately predict conversion risk. Because modifiable vascular and lifestyle factors may influence brain structure, future work should evaluate whether integrating such variables with imaging biomarkers improves individualized risk estimation and informs preventive strategies. Although our cohort is among the largest available with standardized imaging and follow-up, independent validation in external RIS datasets is required to confirm generalizability. Future multi-center studies will test the stability of clustering and risk estimates across populations and scanners.

In conclusion, our study demonstrates the potential of unsupervised learning MRI analysis to predict RIS conversion to clinical MS. By identifying brain aging patterns and structural changes, we propose that BAG could be a novel biomarker for early identification of high-risk RIS individuals, enabling more tailored monitoring and treatment. Nevertheless, these findings suggest that BAG and manifold-derived clustering provide complementary structural information rather than replacing established clinical predictors.

BrainGML-MS integrates BAG and other surrogate biomarkers used in clinical practice to enhance model precision in predicting clinical MS risk. This approach could improve clinical decision-making by distinguishing those who may benefit from early intervention from those suitable for conservative management. Prospective validation studies are required before clinical implementation. Future work should evaluate whether such models improve decision-making, patient outcomes, or cost-effectiveness in real-world settings.

## Materials and methods

### Ethics approval

All healthy datasets were collected under Institutional Review Board (IRB) approval from the French Society of Radiology (Ethics Committee of Research) with the number CRM-2302–320. The datasets were de-identified before model development. Because the study was retrospective, the IRB of the French Society of Radiology (https://cerf.radiologie.fr/cerim) waived the requirement for informed consent. All experiments were conducted in accordance with relevant guidelines and regulations, including the Declaration of Helsinki. The study for RIS and MS patients was approved by MR004 2023-BS-586 at Nice University Hospital.

### Study subjects

**Patient selection.** RIS individuals who fulfilled the 2023 RIS criteria [3] were selected from the RISC database (NCT05388331). The inclusion criterion for this study requires that the MR protocol include a 3D T1-weighted brain scan on a 3T MRI. Demographic data, including age and sex at the RIS index scan, MRI characteristics (including brain lesion location), spinal cord MRI results, and CSF results (if available) were collected from medical records. Information on cardiometabolic and lifestyle factors (e.g., smoking status, diabetes, and hypertension) was not consistently available across participants and, therefore, could not be incorporated into the analyses.

**Healthy control datasets.** The healthy reference dataset included scans acquired at both 1.5T and 3T across 103 sites. This heterogeneity was intentionally introduced to capture scanner-related variability within the normative distribution and to improve the robustness of individualized deviation estimates. We utilized 3D T1-weighted MRI data from nine publicly available databases spanning the entire lifespan. After quality control, we retained 4,359 subjects (male = 2,056; mean age = 32.63, SD = 21.52; female = 2,303; mean age = 37.02, SD = 23.45) from the initial 5,544 subjects. This first dataset was used for brain segmentation and the generation of digital twins. A supplementary dataset of matched controls specific to our study (n = 152, mean age 40, SD = 14; female = 85) was included for statistical analysis.

### Brain Generative Manifold Learning Framework for Multiple Sclerosis (BrainGML-MS): Methodology and application

The intrinsic analysis of the BrainGML-MS framework comprised two successive phases (Fig 5). The initial phase involved segmenting T1-weighted MRIs into 196 cortical and subcortical volumes and calculating asymmetry indices. AssemblyNet was used for brain segmentation in this study because it has been shown to be robust to acquisition variability across scanners. Two recent evaluations published in *Scientific Reports* demonstrated that, among multiple segmentation algorithms compared (AssemblyNet, FastSurfer, FreeSurfer), AssemblyNet exhibited the lowest errors and most consistent

BrainGML®-MS foundation model

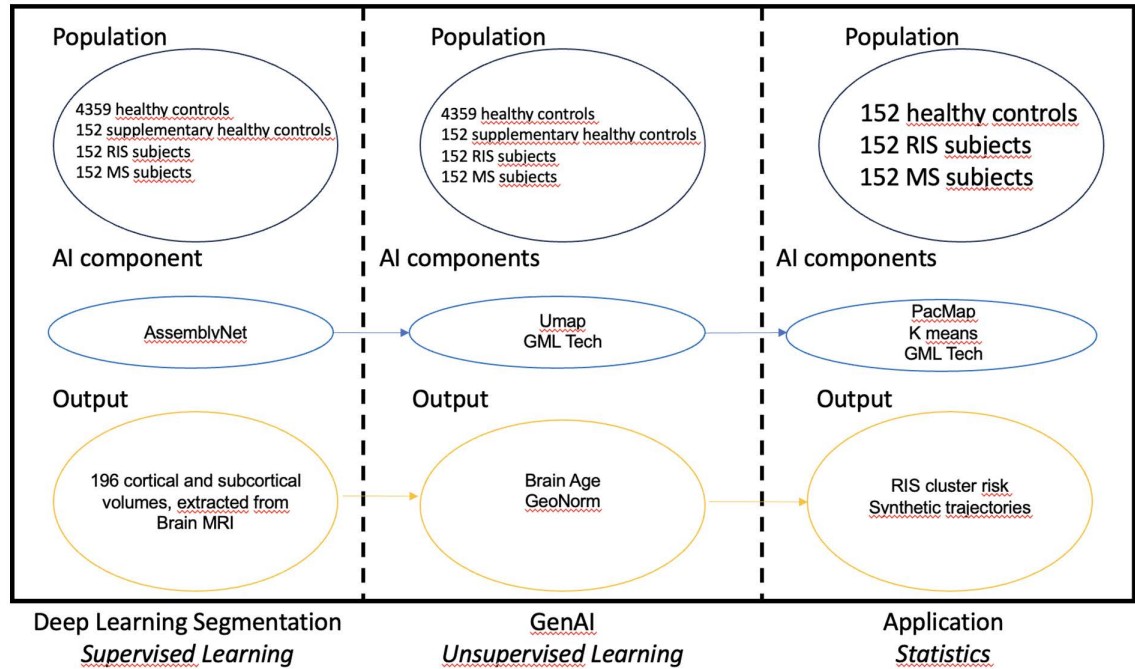

**Fig 5. BrainGML-MS analytical workflow.** Diagram illustrating the two-stage framework used for analysis. Phase 1: Deep learning segmentation of T1-weighted MRI into regional brain volumes and asymmetry indices. Phase 2: Generation of personalized digital twins using manifold learning, followed by brain age estimation, z-score computation, clustering, and trajectory modeling. Output: Individualized structural deviation metrics and conversion risk stratification. The workflow demonstrates how segmentation, digital twin modeling, and manifold analysis are integrated to produce interpretable individualized predictions from MRI data.

volumetric estimates across 1.5 T and 3 T data, indicating reduced sensitivity to magnetic field strength variability and other machine effects in large multi-site datasets [31,32].

 AssemblyNet software produced a deep learning-based segmentation of the entire brain as part of the volBrain pipeline with the following steps: i) denoising, ii) inhomogeneity correction, iii) affine registration into the MNI space, tissue-based intensity normalization, and iv) intracranial cavity volume extraction. Finally, image intensities were centralized and normalized within the brain mask, and the background was set to zero. After preprocessing, the brain was automatically segmented using 250 deep learning models. An unsupervised manifold approximation and projection (UMAP) algorithm was employed to learn population-level patterns [33]. This approach captured joint variations across brain volumes, enabling the unsupervised identification of prominent features in the population. Specifically, when assessing the segmentation of a new patient, it was compared to the 30 nearest neighbors within the reference population to create its digital twin. To determine the personalized normal variations for this new patient, we employed a Leave-One-Out strategy to sequentially estimate the digital twins for each of the 30 HC. We then calculated the standard deviation between the digital twins of the 30 HC and their actual values to establish the local normal variability for the analyzed patient. The patient's brain age was defined as the average age of the 30 nearest individuals in the assumed healthy database. The difference between this estimated brain age and the patient's chronological age was referred to as the brain age gap, a key feature highlighted in the study. BAG reflects deviations in brain structure and serves as a biomarker of accelerated brain aging, often associated with neurodegenerative processes. All the quantitative parameters from the MRI segmentation step with AssemblyNet were used to calculate brain age.

Because the patient's brain age is not normalized and therefore cannot be directly compared across individuals, we used the BAG. Next, we computed the difference between the patient's values and their corresponding digital twin. Then, we compared this to the standard deviation of the controls using a z-score metric called GeoNorm [11], and extracted acceptable value ranges from these z-scores.

Using the z-score dimensionality-reduction algorithm, PacMAP [32] was applied because it effectively balances the preservation of local and global relationships in the reduced dimensions.

## Statistical analysis

**Brain age gap estimation in RIS and other classes:** First, we evaluated chronological age and BAG values across groups to determine whether significant differences existed. Since the BAG values did not follow a Gaussian distribution, a Kruskal-Wallis test was conducted.

The proportions of individuals with RIS-NC and RIS-C were unequal, necessitating a balanced number of subjects across time frames.

**Analysis of brain structures in different groups:** Next, we analyzed differences in brain structures among the groups using z-scores. A z-score below -1.96 or above 1.96 corresponds to a value significantly different at the 5% level. We first assessed the number of regions that showed significant differences between the groups and then performed a Mann-Whitney test. Then, we examined the ten regions (Table 2) that showed the most important differences between groups to identify any distinguishing patterns for RIS conversion. These ten regions correspond to approximately 5% of the total regions of interest (ROI). A column similarity analysis was conducted to identify overlapping brain regions across different conditions.

**Cluster analysis and conversion risk modeling:** K-means clustering was performed on manifold-embedded data of RIS subjects. Cluster-specific conversion rates were calculated and compared using Fisher's exact test. Bootstrap-based 95% confidence intervals were computed for the conversion risk estimates within each cluster. Multinomial logistic regression was used to model cluster membership, incorporating age, sex, BAG, lesion load, and presence of a spinal cord lesion.

**Mapping synthetic trajectories:** The patients' positions along the manifold were transformed into pseudo-temporal scores that represent the geodesic distance to the HC centroid. Generalized additive models were fitted to model the trajectory of key ROI z-scores (e.g., 3rd ventricle, thalamus, cerebellar grey matter) along this progression axis.

All plots were produced using Python and Matplotlib's Plotly library or Seaborn. Data processing was achieved using Sklearn's toolbox. Brain images were drawn using Nilearn. Brain volume ROI was visualized using BrainGlass plotting with Nilearn from the Nipype ecosystem.

## Acknowledgments

The authors thank Mrs. Senia Belkorche for helping with data collection.

## Author contributions

**Conceptualization:** christine lebrun-frenay, Felix Renard, Darin T Okuda, Arnaud Attyé.

**Data curation:** christine lebrun-frenay, Lydiane Mondot, Cassandre Landes-Chateau.

**Formal analysis:** Felix Renard, Lydiane Mondot, Adeline Stewart, Mikael Cohen, Arnaud Attyé.

**Investigation:** christine lebrun-frenay, Felix Renard, Lydiane Mondot, Cassandre Landes-Chateau, Adeline Stewart, Mikael Cohen, Arnaud Attyé.

**Methodology:** christine lebrun-frenay, Felix Renard, Darin T Okuda, Arnaud Attyé.

**Project administration:** christine lebrun-frenay, Arnaud Attyé.

**Resources:** christine lebrun-frenay, Felix Renard, Lydiane Mondot, Cassandre Landes-Chateau, Adeline Stewart, Mikael Cohen.

**Software:** Felix Renard.

**Supervision:** christine lebrun-frenay, Arnaud Attyé.

**Validation:** Lydiane Mondot, Mikael Cohen, Arnaud Attyé.

**Visualization:** Cassandre Landes-Chateau.

**Writing – original draft:** christine lebrun-frenay, Felix Renard, Arnaud Attyé.

**Writing – review & editing:** christine lebrun-frenay, Felix Renard, Lydiane Mondot, Cassandre Landes-Chateau, Adeline Stewart, Mikael Cohen, Darin T Okuda, Arnaud Attyé.

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
