## [Decision Letter · Decision Letter 0]

8 Feb 2026

Response to Reviewers'. This file does not need to include responses to any formatting updates and technical items listed in the 'Journal Requirements' section below.'. This file does not need to include responses to any formatting updates and technical items listed in the 'Journal Requirements' section below.* A marked-up copy of your manuscript that highlights changes made to the original version. You should upload this as a separate file labeled 'Revised Manuscript with Track Changes'.'.* An unmarked version of your revised paper without tracked changes. You should upload this as a separate file labeled 'Manuscript'.'. If you would like to make changes to your financial disclosure, competing interests statement, or data availability statement, please make these updates within the submission form at the time of resubmission. Guidelines for resubmitting your figure files are available below the reviewer comments at the end of this letter. We look forward to receiving your revised manuscript. Kind regards, Liam G McCoyGuest EditorPLOS Digital Health Dhiya Al-Jumeily OBESection EditorPLOS Digital Health Leo Anthony CeliEditor-in-ChiefPLOS Digital Healthorcid.org/0000-0001-6712-6626 **Journal Requirements:** 1. We do not publish any copyright or trademark symbols that usually accompany proprietary names, eg (R), (C), or TM  (e.g. next to drug or reagent names). Please remove all instances of trademark/copyright symbols throughout the text, including BrainGML®-MS.  If the reviewer comments include a recommendation to cite specific previously published works, please review and evaluate these publications to determine whether they are relevant and should be cited. There is no requirement to cite these works unless the editor has indicated otherwise.  **Additional Editor Comments (if provided):** Dear Dr. Lebrun-Frenay,

Thank you for submitting your manuscript to PLOS Digital Health. Your study evaluating the BrainGML-MS software for stratifying individuals with radiologically isolated syndrome (RIS) at higher risk of conversion to clinical multiple sclerosis addresses an important clinical need and applies a novel unsupervised AI approach to this challenging prognostic question.

Your manuscript has been evaluated by two expert reviewers with expertise in multiple sclerosis, neuroimaging, and machine learning approaches. Both reviewers recognized the scientific merit and novelty of your approach, noting the strength of including healthy controls and established MS patients as comparators to the RIS cohort. However, they have raised several concerns that require your attention before we can proceed with publication.

Clinical and methodological characterization: Reviewers requested additional clarity on the clinical features of your cohort, including whether converters developed RRMS versus PPMS, the potential impact of mixed field strengths (1.5T and 3T) on volumetric measurements, and whether cardiometabolic comorbidities—which could independently influence brain age gap—were documented. Please also confirm explicitly that RIS individuals met the 2023 diagnostic criteria.

Generalizability and clinical utility: A central concern is that the model has not been validated on an independent cohort. Additionally, reviewers questioned whether the identified clusters provide meaningful risk stratification beyond established predictors (age, spinal cord lesions, CSF findings), given that the observed conversion rates overlap with previously published estimates. The discussion of clinical implementation would benefit from a more measured perspective on the distance between these findings and bedside application.

Reproducibility and transparency. We note that while the underlying segmentation tool (AssemblyNet) is publicly available, the BrainGML-MS algorithm itself is proprietary. For a study whose primary contribution is a novel AI-driven prognostic tool, this presents a significant barrier to independent replication and external validation by other research groups. We ask the authors to address this limitation explicitly in the revised manuscript and, where possible, provide additional methodological detail sufficient to allow readers to evaluate the approach critically. If regulatory or commercial considerations preclude full code release, the authors should discuss what steps might enable future independent validation (e.g., collaborative access arrangements, validation APIs, or plans for eventual open-source release). Similarly, further explanations of the underlying mechanisms can be made even in the absence of full code release.

Please address these points in your revised manuscript and provide a response letter detailing the changes made. We anticipate these revisions can be accomplished without additional data collection.

**Reviewers' Comments:** Reviewer's Responses to Questions

**Comments to the Author**

1. Does this manuscript meet PLOS Digital Health’s publication criteria? Is the manuscript technically sound, and do the data support the conclusions? The manuscript must describe methodologically and ethically rigorous research with conclusions that are appropriately drawn based on the data presented.? Is the manuscript technically sound, and do the data support the conclusions? The manuscript must describe methodologically and ethically rigorous research with conclusions that are appropriately drawn based on the data presented.

Reviewer #1: Yes

Reviewer #2: Partly

2. Has the statistical analysis been performed appropriately and rigorously?

Reviewer #1: Yes

Reviewer #2: Yes

3. Have the authors made all data underlying the findings in their manuscript fully available (please refer to the Data Availability Statement at the start of the manuscript PDF file)?

The PLOS Data policy requires authors to make all data underlying the findings described in their manuscript fully available without restriction, with rare exception. The data should be provided as part of the manuscript or its supporting information, or deposited to a public repository. For example, in addition to summary statistics, the data points behind means, medians and variance measures should be available. If there are restrictions on publicly sharing data—e.g. participant privacy or use of data from a third party—those must be specified.requires authors to make all data underlying the findings described in their manuscript fully available without restriction, with rare exception. The data should be provided as part of the manuscript or its supporting information, or deposited to a public repository. For example, in addition to summary statistics, the data points behind means, medians and variance measures should be available. If there are restrictions on publicly sharing data—e.g. participant privacy or use of data from a third party—those must be specified.

Reviewer #1: Yes

Reviewer #2: Yes

4. Is the manuscript presented in an intelligible fashion and written in standard English?

Reviewer #1: Yes

Reviewer #2: Yes

Reviewer #1: This manuscript uses a new unsupervised AI-driven model (BrainGML-MS) to determine if factors could be found that could determine if persons with radiologically isolated syndrome/asymptomatic or presymptomatic multiple sclerosis (MS) were at higher risk of conversion to having clinical manifestations of MS. This is an important study as although risk factors have been identified for RIS converting to clinical MS, these are not definitive and so if other factors could be uncovered with a relatively unbiased model, if you will, it would be helpful. Volumetric differences were seen between RIS-C and RIS-NC with RIS-C having atrophy of structures more similar to MS, with additional impact of the presence of spinal cord lesions. One of the strengths of the study is inclusion of healthy controls and those already with clinical manifestations of MS as comparators to RIS. I have a few comments seeking clarification for certain aspects of the manuscript.

Major Comments:

1. Did all the RIS-C participants develop RRMS or did some develop PPMS? If RIS-C participants developed RRMS and PPMS, was there anything additional in the model that was indicative of participants developing RRMS vs. PPMS? One would anticipate spinal cord lesions would be a factor given this has been previously shown by your RISC (Kantarci et al., Ann Neurol 2016), but this is already raised by your model in general for RIS-C vs. RIS-NC.

2. Was there any impact of magnetic field strength on the results, given data acquired at both 1.5T and 3T was acquired? Was 1.5T felt to be sufficient for volumetrics with the model?

3. Is there any data on cardiometabolic comorbidities (hypertension, hyperlipidemia, type 2 diabetes, smoking) for the participants? This could influence the brain age gap and so if this information is available, would be helpful to include given that differences between many group comparisons were significant as well. Cardiometabolic comorbidities are potentially reversible, so if present in the RIS-C group, could also be helpful to include as part of management to try to prevent conversion to MS.

Minor Comment:

1. Presumably all RIS individuals met 2023 RIS criteria? If so, this could be stated in the Patient Selection section of the Materials and Methods.

Reviewer #2: In this paper the Authors evaluated whether a new unsupervised AI-driven model could stratify

individuals with RIS at a greater risk of conversion. A total of 152 RIS individuals that will evolve to clinical MS or not during the follow-up were compared to 152 MS patients and 152 healthy controls. The unsupervised analysis identified four RIS clusters with varying conversion risks, ranging from 10% to 39% over five years. Brain aging patterns and the presence of spinal cord lesions differed significantly among groups.

The approach taken in this paper is of clear scientific interest and novelty. I found, however, the following issues:

-- a concern about generalizability of the results, as the proposed clustering and risk stratification are not validated on an independent cohort of RIS subjects;

-- to my knowledge the BrainGML®-MS algorithm is not publicly available, which is an obstacle to independent replication of the data or assessment of its methodological characteristics.

-- it remains insufficiently demonstrated whether BAG and cluster outperform or add to other already known strong predictors of evolution (e.g., age, spinal cord lesions, CSF OCBs…).

-- conversion probabilities (10–39% at 5 years) overlap substantially with previously reported RIS risks, Therefore, does the model really provide clinically useful stratification beyond existing risk models?.

--While the discussion suggests implications for early treatment decisions, clinical implementation appears rather speculative. This issue should be discussed using a more critical approach.

**Do you want your identity to be public for this peer review?** For information about this choice, including consent withdrawal, please see our Privacy Policy..

Reviewer #1: No

Reviewer #2: No

**Figure resubmission:** While revising your submission, we strongly recommend that you use PLOS’s NAAS tool (https://ngplosjournals.pagemajik.ai/artanalysis) to test your figure files. NAAS can convert your figure files to the TIFF file type and meet basic requirements (such as print size, resolution), or provide you with a report on issues that do not meet our requirements and that NAAS cannot fix.

**Reproducibility:** To enhance the reproducibility of your results, we recommend that authors of applicable studies deposit laboratory protocols in protocols.io, where a protocol can be assigned its own identifier (DOI) such that it can be cited independently in the future. Additionally, PLOS ONE offers an option to publish peer-reviewed clinical study protocols. Read more information on sharing protocols at https://plos.org/protocols?utm_medium=editorial-email&utm_source=authorletters&utm_campaign=protocols To enhance the reproducibility of your results, we recommend that authors of applicable studies deposit laboratory protocols in protocols.io, where a protocol can be assigned its own identifier (DOI) such that it can be cited independently in the future. Additionally, PLOS ONE offers an option to publish peer-reviewed clinical study protocols. Read more information on sharing protocols at https://plos.org/protocols?utm_medium=editorial-email&utm_source=authorletters&utm_campaign=protocols

---

## [Decision Letter · Decision Letter 1]

3 Apr 2026

Predicting multiple sclerosis from radiologically isolated syndrome using generative artificial intelligence

PDIG-D-25-00290R1

Dear prof lebrun-frenay,

We are pleased to inform you that your manuscript 'Predicting multiple sclerosis from radiologically isolated syndrome using generative artificial intelligence' has been provisionally accepted for publication in PLOS Digital Health.

Best regards,

Liam G McCoy, MD, MSc

Guest Editor

PLOS Digital Health

**Additional Editor Comments (if provided):**

The authors have appropriately addressed all reviewer comments, and have made reasonable steps toward openness and reproducibility.

**Reviewer Comments (if any, and for reference):**

Reviewer's Responses to Questions

**Comments to the Author**

Reviewer #1: All comments have been addressed

Reviewer #2: All comments have been addressed

publication criteria? Is the manuscript technically sound, and do the data support the conclusions? The manuscript must describe methodologically and ethically rigorous research with conclusions that are appropriately drawn based on the data presented.? Is the manuscript technically sound, and do the data support the conclusions? The manuscript must describe methodologically and ethically rigorous research with conclusions that are appropriately drawn based on the data presented.

Reviewer #1: Yes

Reviewer #2: Yes

3. Has the statistical analysis been performed appropriately and rigorously?

Reviewer #1: Yes

Reviewer #2: Yes

4. Have the authors made all data underlying the findings in their manuscript fully available (please refer to the Data Availability Statement at the start of the manuscript PDF file)?

The PLOS Data policy requires authors to make all data underlying the findings described in their manuscript fully available without restriction, with rare exception. The data should be provided as part of the manuscript or its supporting information, or deposited to a public repository. For example, in addition to summary statistics, the data points behind means, medians and variance measures should be available. If there are restrictions on publicly sharing data—e.g. participant privacy or use of data from a third party—those must be specified.requires authors to make all data underlying the findings described in their manuscript fully available without restriction, with rare exception. The data should be provided as part of the manuscript or its supporting information, or deposited to a public repository. For example, in addition to summary statistics, the data points behind means, medians and variance measures should be available. If there are restrictions on publicly sharing data—e.g. participant privacy or use of data from a third party—those must be specified.

Reviewer #1: Yes

Reviewer #2: Yes

5. Is the manuscript presented in an intelligible fashion and written in standard English?

Reviewer #1: Yes

Reviewer #2: Yes

Reviewer #1: All comments have been addressed.

Reviewer #2: The Authors have adequately addressed the reviewers' comments

**Do you want your identity to be public for this peer review?** For information about this choice, including consent withdrawal, please see our Privacy Policy..

Reviewer #1: No

Reviewer #2: No
